# Joint Entity and Relation Extraction with Span Pruning and Hypergraph Neural Networks

**Zhaohui Yan**[1,2], **Songlin Yang**[3*], **Wei Liu**[1,2], **Kewei Tu**[1,2†]

[1]School of Information Science and Technology, ShanghaiTech University
[2]Shanghai Engineering Research Center of Intelligent Vision and Imaging
[3]MIT CSAIL

{yanzhh, liuwei4, tukw}@shanghaitech.edu.cn
yangsl66@mit.edu

## Abstract

Entity and Relation Extraction (ERE) is an important task in information extraction. Recent marker-based pipeline models achieve state-of-the-art performance, but still suffer from the error propagation issue. Also, most of current ERE models do not take into account higher-order interactions between multiple entities and relations, while higher-order modeling could be beneficial.In this work, we propose Hyper-Graph neural network for ERE (HGERE), which is built upon the PL-marker (a state-of-the-art marker-based pipleline model). To alleviate error propagation,we use a high-recall pruner mechanism to transfer the burden of entity identification and labeling from the NER module to the joint module of our model. For higher-order modeling, we build a hypergraph, where nodes are entities (provided by the span pruner) and relations thereof, and hyperedges encode interactions between two different relations or between a relation and its associated subject and object entities. We then run a hypergraph neural network for higher-order inference by applying message passing over the built hypergraph. Experiments on three widely used benchmarks (ACE2004, ACE2005 and SciERC) for ERE task show significant improvements over the previous state-of-the-art PL-marker. [1]

## 1 Introduction

Entity and Relation Extraction (ERE) is a fundamental task in information extraction (IE), compromising two sub-tasks: Named Entity Recognition (NER) and Relation Extraction (RE). There is a long debate on joint vs. pipeline methods for ERE. Pipeline decoding extracts entities first and predicts relations solely on pairs of extracted entities, while joint decoding predicts entities and relations simultaneously.

Recently, the seminal work of (Zhong and Chen, 2021) shows that pipeline decoding with a frustratingly simple marker-based encoding strategy — i.e., inserting solid markers (Baldini Soares et al., 2019; Xiao et al., 2020) around predicted subject and object spans in the input text — achieves state-of-the-art RE performance. Modified sentences (with markers) are fed into powerful pretrained large language models (LLM) to obtain more subject- and object-aware representations for RE classification, which is the key to the performance improvement. However, current marker-based pipeline models (e.g., the recent state-of-the-art ERE model PL-marker (Ye et al., 2022)) *only* send predicted entities from the NER module to the RE module, therefore missing entities would never have the chance to be re-predicted, suffering from the *error propagation issue*. On the other hand, for joint decoding approaches (e.g. Table Filling methods (Miwa and Sasaki, 2014; Zhang et al., 2017; Wang and Lu, 2020))—though they do not suffer from the error propagation issue—it is hard to incorporate markers for leveraging LLMs, since entities are not predicted prior to relations. Our desire is to obtain the best of two worlds, being able to use marker-based encoding mechanism for enhancing RE performance and meanwhile alleviating the error propagation problem. We adopt PL-marker as the backbone of our proposed model and a span pruning strategy to mitigate error propagation. That is, instead of sending only predicted entity spans to the RE module, we *over-predict* candidate spans so that the recall of gold entity spans is nearly perfect (but there also could be many non-entity spans), transferring the burden of entity classification and labeling from the NER module to the RE module of PL-marker. The number of over-predicted spans is upper-bounded, balancing the computational complexity of marker-based encoding and the recall of gold entity span. Empirically, we find this simple strategy by itself clearly improves PL-marker.

---

*This work was done when Songlin was at ShanghaiTech.
†Corresponding Author

[1]Source code is availabel at https://github.com/yanzhh/HGERE

We further incorporate a *higher-order* interaction module into our model. Most previous ERE models either implicitly model the interactions between instances by shared parameters (Wang and Lu, 2020; Yan et al., 2021; Wang et al., 2021) or use a traditional graph neural network that models pairwise connections between a relation and an entity (Sun et al., 2019). It is difficult for these approaches to explicitly model higher-order relationships among multi-instances, e.g. the dependency among a relation and its corresponding subject and object entities. Many recent works in structured prediction tasks show that explicit higher-order modeling is still beneficial even with powerful large pretrained encoders (Zhang et al., 2020a; Li et al., 2020; Yang and Tu, 2022; Zhou et al., 2022, *inter alia*), motivating us to use an additional higher-order module to enhance performance.

A common higher-order modeling approach is by means of probabilistic modeling (i.e., conditional random field (CRF)) with end-to-end Mean-Field Variational Inference (MFVI), which can be seamlessly integrated into neural networks as a recurrent neural network layer (Zheng et al., 2015a), and has been widely used in various structured prediction tasks, such as dependency parsing (Wang et al., 2019), semantic role labeling (Li et al., 2020; Zhou et al., 2022), and information extraction (Jia et al., 2022). However, the limitations of CRF modeling with MFVI are i): CRF's potential functions are parameterized in log-linear forms with strong independence assumptions, suffering from low model capacities (Qu et al., 2022), ii) MFVI uses fully-factorized Bernoulli distributions to approximate the otherwise multimodal true posterior distributions, oversimplifying the inference problem and thus is sub-optimal. Therefore we need more *expressive* tools to improve the quality of higher-order inference. Fortunately, there are many recent works in the machine learning community showing that graph neural networks (GNN) can be used as an inference tool and outperform approximate statistical inference algorithms (e.g., MFVI) (Yoon et al., 2018; Zhang et al., 2020b; Kuck et al., 2020; Satorras and Welling, 2021) (see (Hua, 2022) for a survey). Inspired by these works, we employ a hypergraph neural network (HyperGNN) instead of MFVI for high-order inference and propose our model HGERE (HyperGraph Neural Network for ERE). Concretely, we build a hypergraph where nodes are candidate subjects and objects (obtained

from the span pruner) and relations thereof, and hyperedges encode the interactions between either two relations with shared entities or a relation and its associated subject and object entity spans. In contrast, existing GNN models for IE (Sun et al., 2019; Nguyen et al., 2021) only model the pairwise interactions between a relation and one of its corrsponding entity. We empirically show the advantages of our higher-order interaction module (i.e., hypergraph neural network) over MFVI and tranditional GNN models.

Our contribution is three-fold: i) We adopt a simple and effective span pruning method to mitigate the error propagation issue, enforcing the power of marker-based encoding. ii) We propose a novel hypergraph neural network enhanced higher-order model, outperforming higher-order CRF-based models with MFVI. iii) We show great improvements over the prior state-of-the-art PL-marker on three commonly used benchmarks for ERE: ACE2004, ACE2005 and SciERC.

## 2 Background

### 2.1 Problem formulation

Given a sentence $X$ with $n$ tokens: $x_1, x_2, ..., x_n$, an entity span is a sequence of tokens labeled with an entity type and a relation is an entity span pair labeled with a relation type. We denote the set of all entity spans of the sentence with a span length limit $L$ by $S(X) = \{s_1, s_2, ..., s_m\}$ and define $\text{ST}(i)$ and $\text{ED}(i)$ as the start and end token indices of the span $s_i$.

The joint ERE task is to simultaneously solve the NER and RE tasks. Let $\mathcal{C}_e$ be the set of entity types and $\mathcal{C}_r$ be the set of relation types. For each span $s_i \in S(X)$, the NER task is to predict an entity type $y_e(s_i) \in \mathcal{C}_e$ or $y_e(s_i) = \text{null}$ if the span $s_i$ is not an entity. The RE task is to predict a relation type $y_r(r_{ij}) \in \mathcal{C}_r$ or $y_r(r_{ij}) = \text{null}$ for each span $r_{ij} = (s_i, s_j), s_i, s_j \in S(X)$.

### 2.2 Packed levitated marker (PL-marker)

Zhong and Chen (2021) insert two pairs of solid markers (i.e., $[S]$ and $[\backslash S]$) to highlight both the subject and object entity spans in a given sentence, and this simple approach achieves state-of-the-art RE performance. We posit that this is because LLM is more aware of the subject and object spans (with markers) and thus can produce better span representations to improve RE. But this strategy needs to iterate over all possible entity span pairs and is

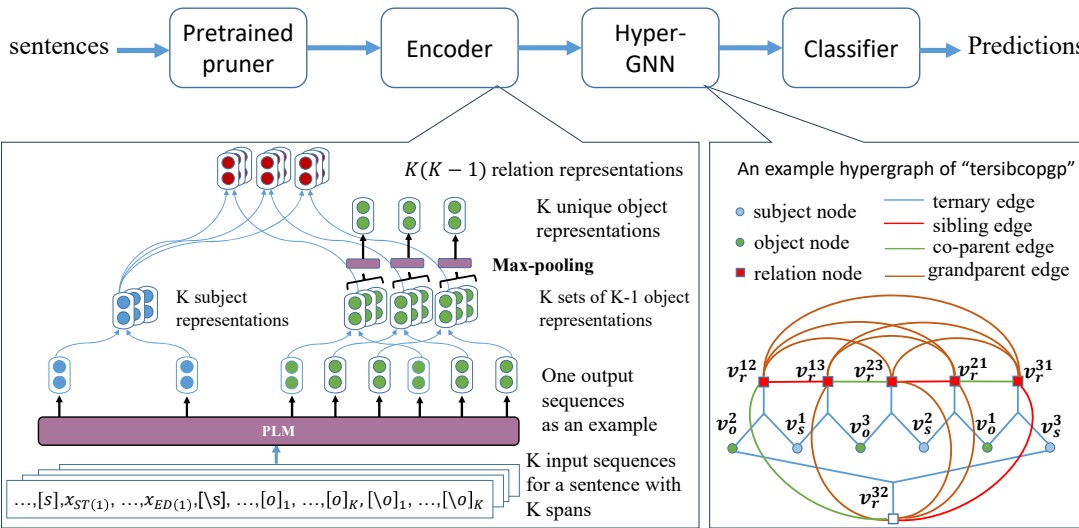

Figure 1: Illustration of our framework

thus slow. To tackle the efficiency problem, Zhong and Chen (2021) propose an approximated variant wherein each possible entity span is associated with a pair of levitated markers (i.e., $[O]$ and $[\backslash O]$) whose representations are initialized with the positional embedding of the span's start and end tokens, and all such levitated markers are concatenated to the end of the sentence. As such, levitated-marker-based encoding needs only one pass, significantly improving efficiency at the cost of slight performance drop. Zhong and Chen (2021) also propose a masked attention mechanism such that the original input text tokens are not able to attend to markers, while markers can attend to paired markers (but not unpaired markers) and all input text tokens. As a consequence, the relative positions of levitated markers in the concatenated sentence do not matter at all, eliminating potential implausible inductive bias on the concatenation order.

However, marker-base encoding is only used in RE, not in NER. To leverage marker-based encoding in the NER module for modeling span inter-relations, PL-marker (Ye et al., 2022) associates each possible span with two levitated markers and concatenates all of them to the end of the input sentence. However, this strategy could make the input sentence extremely long since there are quadratic number of spans. To solve this issue, PL-marker clusters the markers based on the starting position of their corresponding spans, and divides them into $N$ groups. Then the input sentence is duplicated $N$ times and each group of levitated markers is concatenated to the end of one sentence copy. Ye et al. (2022) refers to this strategy as *neighborhood-oriented packing scheme*. Furthermore, to balance the efficiency and the model expressiveness, Ye et al. (2022) combine solid markers and levitated markers, proposing *Subject-oriented Packing for Span Pair* in the RE module. That is, if there are $m$ entities, they copy the sentence for $m$ times, and for each copy, they use solid markers to mark a different entity as the subject and concatenate the levitated markers of all other entities (as objects) at the end of the sentence.

## 3 Method

**Overview.** Our method is built upon the state-of-the-art PL-marker. We employ a high-recall span pruner to obtain candidate entity spans, similar to the NER module in PL-marker. However, instead of aiming to accurately predict all possible entity spans, our pruner focuses on removing unlikely candidates to achieve a much higher recall. Then we feed the candidate span set to the RE module to obtain entity and relation representations, which are used to initialize the node representations of our hypergraph neural network for higher-order inference with a message passing scheme. Finally, we perform NER and RE based on the refined entity and relation representations. Fig. 1 depicts the neural architecture of our model.

### 3.1 Span Pruner

We adopt the neighborhood-oriented packing scheme from PL-marker for span encoding, except that we simply predict entity existence (i.e., binary classification) instead of predicting entity labels during the training phrase. See Appendix A.4 for

details.

To produce a candidate span set, we rank all the spans by their scores and take top $K$ as our prediction $S_p(X)$. We assume that the number of entity spans of a sentence is linear to its length $n$, so $K$ is set to $\lambda \cdot n$ where $\lambda$ is a coefficient. For a very long sentence, the number of entity spans is often sublinear to $n$, while for a very short sentence, we wish to keep enough candidate spans, so we additionally set an upper and lower bound: $K = \max(l_{\min}, \min(\lambda \cdot n, l_{\max}))$.

In practice, with our span pruner, more than 99% gold entity spans are included in the candidate set for all three datasets. If we predict entities as in PL-marker instead of pruning, only around 95% and 80% gold entities are kept in the predicted entities for ACE2005 and SciERC respectively, leading to severe error propagation (see §5.1 for an ablation study).

The span pruner is trained independently from the joint ERE model introduced in the next section. This is because the joint ERE training loss will be defined based on candidate entity spans produced by the span pruner. When sharing parameters, the pruner would provide a different candidate span set during training, leading to moving targets and thereby destabilizing the whole training process.

## 3.2 Joint ERE Model: First-order Backbone

The backbone module is based on the RE module of PL-marker. Concretely, given an input sentence $X = \{x_1, x_2, ..., x_n\}$ and a subject span $s_i = (x_{\text{ST}(i)}, x_{\text{ED}(i)}) \in S_p(X)$ provided by the span pruner, every entity span $s_j \in S_p(X), 1 \leq j \leq K, j \neq i$ could be a candidate object span of $s_i$. The module inserts a pair of solid markers $[S]$ and $[\backslash S]$ before and after the subject span and assign every object span $s_j$ a pair of levitated markers $[O]_j$ and $[\backslash O]_j$. As shown below, the levitated markers are packed together and inserted at the end of the input sequence to a PLM:

$$x_1, ..., [S], x_{\text{ST}(i)}, ..., x_{\text{ED}(i)}, [\backslash S], ..., x_n,$$
$$[O]_1, ..., [O]_K, [\backslash O]_1, ..., [\backslash O]_K$$

Then we obtain the contextualized hidden representation $\mathbf{h}_x$ of the modified input sequence and the final subject representation is:

$$\mathbf{h}_s(s_i) = \text{FFN}_s([\mathbf{h}_x([S]); \mathbf{h}_x([\backslash S])])$$

FFN represents a single linear layer in this work. The object representation of $s_j$ for the current subject $s_i$ and the representation of relation $r_{ij} =$

$(s_i, s_j)$ are:

$$\mathbf{h}_o^i(s_j) = \text{FFN}_o([\mathbf{h}_x([O]_j); \mathbf{h}_x([\backslash O]_j)])$$

$$\mathbf{h}_r(r_{ij}) = \text{FFN}_r([\mathbf{h}_s(s_i); \mathbf{h}_o^i(s_j)])$$

Repeating $K$ times, we get all $K$ subject representations and $K(K-1)$ relation representations. As the object representation of $s_j$ is not identical for different subject span $s_i$, there are $K$ object representation sets $\mathbf{h}_o^i, 1 \leq i \leq K$. We apply a max-pooling layer to obtain a unique object representation for each object span $s_j \in S_p(X)$:

$$\mathbf{h}_o(s_j) = \text{Maxpooling}_{1 \leq i \leq K, i \neq j}(\mathbf{h}_o^i(s_j))$$

## 3.3 Joint ERE Model: Higher-order Inference with Hypergraph Neural Networks

**Hypergraph Building** So far, the representations of the entities and relations from the backbone module do not explicitly consider beneficial interactions among related instances. To model higher-order interactions among a relation and its associated subject and object entities as well as between any two relations sharing an entity, we build a hypergraph $\mathcal{G} = (\mathcal{V}, \mathcal{E})$ to connect the related instances. The nodes set $\mathcal{V}$ is composed of candidate subjects, objects (provided by the span pruner) and all possible pairwise relations thereof, and we denote them as $\mathcal{V}_s = \{v_s^i | i \in [1, K]\}$, $\mathcal{V}_o = \{v_s^j | j \in [1, K]\}$ and $\mathcal{V}_r = \{v_r^{ij} | i, j \in [1, K], i \neq j\}$.

Hyperedges $\mathcal{E}$ capture the interactions we are concerned with, and they can be divided into two categories: the subject-object-relation (sub-obj-rel) hyperedges $\mathcal{E}_{sor}$ and the relation-relation (rel-rel) hyperedges $\mathcal{E}_{rr}$. Each hyperedge $e_{sor}^{ij} \in \mathcal{E}_{sor}$ connects a subject node $v_s^i$, an object node $v_o^j$ and the corresponding relation node $v_r^{ij}$, and we refer to these hyperedges as *ternary* edges (*ter* for short). Each rel-rel edge $e_{rr}^{ijk} \in \mathcal{E}_{rr}$ connects two relation nodes with a shared subject or object entity. We assume in a relation, the subject is the parent node and the object is the child node, and then we can refine rel-rel edges into three subtypes, *sibling* (*sib*, connecting $v_r^{ij}$ and $v_r^{ik}$), *co-parent* (*cop*, connecting $v_r^{ij}$ and $v_r^{kj}$) and *grand-parent* (*gp*, connecting $v_r^{ij}$ and $v_r^{jk}$), following the common definitions in the dependency parsing literature.

If we incorporate all aforementioned hyperedges into the hypergraph, we obtain the *tersibcopgp* variant which is illustrated in Fig. 1. By removing some types of hyperedges we can get different variants, but without loss of generality we describe

the message passing scheme in the following using *tersibcopgp*.

As such, we can define a CRF on the hypergraph and leverage probabilistic inference algorithms such as MFVI for higher-order inference. However, as discussed in §1, we can use a more expressive method to improve inference quality and introduce a HyperGraph Neural Network (HGNN) as described next.

**Initial node representation**  For a relation node $v_r^{ij}$ with its associated subject node $v_s^i$ and object node $v_o^j$, we use $\mathbf{g}^l(v_r^{ij}), \mathbf{g}^l(v_s^i), \mathbf{g}^l(v_o^j)$ to denote their respective representation outputs from the $l$-th HGNN layer. Initial node representations (before being fed to a HGNN) are $\mathbf{g}^0(v_s^i) = \mathbf{h}_s(s_i)$, $\mathbf{g}^0(v_o^j) = \mathbf{h}_o(s_j)$ and $\mathbf{g}^0(v_r^{ij}) = \mathbf{h}_r(r_{ij})$, respectively (from the backbone module).

**Message representation**  A hyperedge connecting to nodes serve as the bridge for message passing between nodes connected by it. Let $\mathcal{N}_e(v)$ be the set of hyperedges connecting to a node $v$.

For a *ter* hyperedge $e_{ter}^{ij} \in \mathcal{E}_{sor}$ connecting a subject node $v_s^i$, a object node $v_o^j$ and a relation node $v_r^{ij}$, the message representation it carries is:

$$\mathbf{hr}_{ij}^l = \text{FFN}_r^{ter}(\mathbf{g}^{l-1}(v_r^{ij}))$$
$$\mathbf{hs}_i^l = \text{FFN}_s^{ter}(\mathbf{g}^{l-1}(v_s^i))$$
$$\mathbf{ho}_j^l = \text{FFN}_o^{ter}(\mathbf{g}^{l-1}(v_o^j))$$
$$\mathbf{m}^l(e_{ter}^{ij}) = \text{FFN}_e^{ter}(\mathbf{hr}_{ij}^l \circ \mathbf{hs}_i^l \circ \mathbf{ho}_j^l)$$

where $\circ$ is the Hadamard product.

A rel-rel edge $e_z^{ijk} \in \mathcal{E}_{rr}, z \in \{sib, cop, gp\}$ connects two relations sharing an entity. For simplicity, we denote them relation $a$ and $b$. If we fix $a$ as $a \triangleq v_r^{ij}$, then as previously described, relation $b$ is $v_r^{ik}$ for *sib* edge, $v_r^{kj}$ for *cop* edge, and $v_r^{jk}$ for *gp* edge. The message $e_z^{ijk}$ carries is given by,

$$\mathbf{h}_z^l(a) = \text{FFN}_a^z(\mathbf{g}^{l-1}(a))$$
$$\mathbf{h}_z^l(b) = \text{FFN}_b^z(\mathbf{g}^{l-1}(b))$$
$$\mathbf{m}^l(e_z^{ijk}) = \text{FFN}_e^z(\mathbf{h}_z^l(a) \circ \mathbf{h}_z^l(b))$$

**Node representation update**  We aggregate messages for each node $v \in \mathcal{V}$ from adjacent edges $\mathcal{N}_e(v)$ with an attention mechanism by taking a learned weighted sum, and add the aggregated message to the prior node representation,

$$\beta^l(e, v) = \mathbf{w}^\top \sigma(\mathbf{W}[\mathbf{g}^{l-1}(v); \mathbf{m}^l(e)]$$
$$\alpha^l(e, v) = \frac{\exp \beta^l(e, v)}{\sum_{e' \in \mathcal{N}_e(v)} \exp \beta^l(e', v)}$$
$$\mathbf{g}^l(v) = \mathbf{g}^{l-1}(v) + \sum_{e \in N_e(v)} \alpha^l(e, v)\mathbf{m}^l(e)$$

where $\sigma(\cdot)$ is a non-linear activator and $\mathbf{w}, \mathbf{W}$ are two trainable parameters. An entity node would receive messages only from *ter* edges while a relation node would receive messages from both *ter* edges and rel-rel edges.

**Training**  We obtain refined $\mathbf{g}^l(v)$ from the final layer of HGNN. Give an entity span $s_i \in S_p(X)$, we concatenate the corresponding subject representation $\mathbf{g}^l(v_s^i)$ and object representation $\mathbf{g}^l(v_o^i)$ to obtain the entity representation, and compute the probability distribution over the types $\{\mathcal{C}_e\} \bigcup \{\text{null}\}$:

$$P_e(\hat{y}_e|s_i) = \text{Softmax}(\text{FFN}_e^{cls}([\mathbf{g}^l(v_s^i); \mathbf{g}^l(v_o^i)]))$$

Given a relation $r_{ij} = (s_i, s_j), s_i, s_j \in S_p(X)$, we compute the probability distribution over the types $\{\mathcal{C}_r\} \bigcup \{\text{null}\}$:

$$P_r(\hat{y}_r|r_{ij}) = \text{Softmax}(\text{FFN}_r^{cls}(\mathbf{g}^l(v_r^{ij})))$$

We use the cross-entropy loss for both entity and relation prediction:

$$L_e = - \sum_{s_i \in S_p(X)} \log(P_e(y_e^*(s_i)|s_i))$$
$$L_r = - \sum_{s_i, s_j \in S_p(X)} \log(P_r((y_r^*(r_{ij})|r_{ij}))$$

where $y_e^*$ and $y_r^*$ are gold entity and relation types respectively. The total loss is $L = L_e + L_r$.

## 4   Experiment

**Datasets**  We experiment on SciERC (Luan et al., 2018), ACE2004 (Doddington et al., 2004) and ACE2005 (Walker et al., 2006). We follow Ye et al. (2022) to split ACE2004 into 5 folds and split ACE2005 and SciERC into train/dev/test sets. See Appendix A.1 for detailed dataset statistics.

**Evaluation metrics**  We report micro labeled F1 measures for NER and RE. For RE, the difference between Rel and Rel+ is that the former requires correct prediction of subject and object entity spans and the relation type between them, while the latter additionally requires correct prediction of subject and object entity types.

---

[2]Ye et al. (2022) count a symmetric relation twice for evaluation which is inconsistent with previous work.

| Models | Encoder | ACE2005 | | | ACE2004 | | | SciERC | | |
|---|---|---|---|---|---|---|---|---|---|---|
| | | Ent | Rel | Rel+ | Ent | Rel | Rel+ | Ent | Rel | Rel+ |
| (Wadden et al., 2019)★ | BERT$_B$/ SciBERT | 88.6 | 63.4 | - | - | - | - | 67.5 | 48.4 | - |
| (Wang et al., 2021)★ | | 88.8 | - | 64.3 | 87.7 | - | 60.0 | 68.4 | - | 36.9 |
| (Zhong and Chen, 2021)★ | | 90.1 | 67.7 | 64.8 | 89.2 | 63.9 | 60.1 | 68.9 | 50.1 | 36.8 |
| (Yan et al., 2021) | | - | - | - | - | - | - | 66.8 | - | 38.4 |
| (Shen et al., 2021)★ | | 87.6 | 66.5 | 62.8 | - | - | - | 70.2 | 52.4 | - |
| (Nguyen et al., 2021) | | 88.9 | 68.9 | - | - | - | - | - | - | - |
| (Ye et al., 2022)★$_{re}$ | | 89.1 | 68.3 | 65.1 | 88.5 | 66.3 | 62.2 | 68.8 | 51.1 | 38.3 |
| Backbone★ | | 90.0 | 69.8 | 66.7 | 89.5 | 66.6 | 62.1 | 71.3 | 52.3 | 40.2 |
| GCN★ | | 90.2 | 69.6 | 66.5 | **90.0** | 67.6 | 63.5 | 74.1 | 54.8 | 42.9 |
| MFVI★ | | 90.2 | 69.7 | 67.1 | 89.7 | 67.4 | 63.4 | 73.3 | 54.7 | 42.5 |
| HGERE★ (our model) | | 90.2 | **70.7** | **67.5** | 89.9 | **68.2** | **64.2** | **74.9** | **55.7** | **43.6** |
| (Liu et al., 2022) | T5$_{3B}$ | 91.3 | 72.7 | 70.5 | - | - | - | - | - | - |
| (Wang and Lu, 2020) | ALBERT | 89.5 | 67.6 | 64.3 | 88.6 | 63.3 | 59.6 | - | - | - |
| (Wang et al., 2021)★ | | 90.2 | - | 66.0 | 89.5 | - | 63.0 | - | - | - |
| (Zhong and Chen, 2021)★ | | 90.9 | 69.4 | 67.0 | 90.3 | 66.1 | 62.2 | - | - | - |
| (Yan et al., 2021) | | 89.0 | - | 66.8 | 89.3 | - | 62.5 | - | - | - |
| (Ye et al., 2022)★$_{re}$ | | 91.3 | 72.5 | 70.5 | 90.5 | 69.3 | 66.1 | - | - | - |
| Backbone★ | | 91.5 | 72.9 | 70.2 | 91.6 | 70.2 | 66.6 | - | - | - |
| GCN★ | | 91.7 | 73.1 | 69.9 | **92.0** | 71.5 | 67.9 | - | - | - |
| MFVI★ | | 91.6 | 72.7 | 70.1 | 89.9 | 68.5 | 65.1 | - | - | - |
| HGERE★ (our model) | | **91.9** | **73.5** | **70.8** | 91.9 | **71.9** | **68.3** | - | - | - |

Table 1: F1 scores and standard deviations on ACE2004, ACE2005 and SciERC. The models marked with ★ leverage cross-sentence information. A model with subscript *re* means we re-evaluate the model with the evaluation method commonly used in other work[2]. Backbone, MFVI and GCN are our baseline models.

**Baseline** Our baseline models include: i) **Backbone**. It is described in Sect. 3.2 and does not contain the higher-order interaction module. ii) **GCN**. It has a similar architecture to Sun et al. (2019); Nguyen et al. (2021) and does not contain higher-order hyperedges. See Appendix A.6 for a detailed description. iii) **MFVI**. It defines a CRF on the same hypergraph as our model and uses MFVI instead of hypergraph neural networks for higher-order inference. See Appendix A.5 for a detailed description.

**Implementation details** For a fair comparison with previous work, we use *bert-base-uncased*(Devlin et al., 2019) and *albert-xxlarge-v1*(Lan et al., 2020) as the base encoders for ACE2004 and ACE2005, *scibert-scivocab-uncased* (Beltagy et al., 2019) as the base encoder for SciERC. GCN and MFVI are also built upon Backbone. The implementation details of experiments are in Appendix A.3.

**Main results** For HGERE, we report the best results among the following variants of hypergraphs with different types of hyperedges: *ter, cop, sib, gp, tersib, tercop, tergp, tersibcop, tersibgp, tercopgp,* and *tersibcopgp*. The best variants of HGERE are *tersibcop* on SciERC and ACE2005 (BERT$_B$); *tersib* on ACE2005 (ALBERT); *tercop*

on ACE2004. For MFVI we use the same variants as used in HGERE.

Table 1 shows the main results. Surprisingly, Backbone outperforms prior approaches in almost all metrics by a large margin (except on ACE2004 with BERT$_B$ and ACE2005 with ALBERT), which we attribute to the reduction of error propagation with a span pruning mechanism. Our proposed model HGERE outperforms almost all baselines in all metrics (except the entity metric on ACE2004), validating that using hyperedges to encode higher-order interactions is effective (compared with GCN) and that using hypergraph neural networks for higher-order modeling and inference is better than CRF-based probabilistic modeling with MFVI. Finally, we remark that HGERE obtains state-of-the-art performances on all the three datasets.

## 5 Analysis

### 5.1 Effectiveness of the span pruner

To study the effectiveness of the span pruner, we replace it with an entity identifier which is the original NER module from PL-marker and is trained only on entity existence. The performance of the span pruner and the entity identifier (denoted by Eid) on entity existence is shown in Table 2. We can observe that if we replace the span pruner with the entity identifier, the recall of gold *unlabeled*

| | | SciERC | | | ACE2005 (BERT$_B$) | | |
|---|---|---|---|---|---|---|---|
| | | P | R | F1 | P | R | F1 |
| | train | 98.8 | 98.2 | 98.5 | 100.0 | 99.9 | 99.9 |
| Eid | dev | 81.0 | 81.6 | 81.3 | 94.7 | 94.6 | 94.7 |
| | test | 80.4 | 78.7 | 79.5 | 95.6 | 95.8 | 95.7 |
| | train | 38.0 | 99.2 | 54.9 | 37.2 | 99.9 | 54.2 |
| Pruner | dev | 38.1 | 99.1 | 55.0 | 36.4 | 99.7 | 53.3 |
| | test | 38.7 | 99.2 | 55.7 | 37.0 | 99.8 | 54.0 |

Table 2: Evaluation results on entity existence of the span pruner vs. an entity identifier.

| SciERC | | Ent | Rel | Rel+ |
|---|---|---|---|---|
| Eid | Backbone | 69.4 | 50.3 | 39.0 |
| | HGERE | 69.4 | 51.5 | 39.5 |
| pruner | Backbone | 71.3 | 52.3 | 40.2 |
| | HGERE | 74.9 | 55.7 | 43.6 |
| ACE2005 (BERT$_B$) | | Ent | Rel | Rel+ |
| Eid | Backbone | 89.5 | 68.3 | 65.3 |
| | HGERE | 89.5 | 68.5 | 66.0 |
| pruner | Backbone | 90.0 | 69.8 | 66.7 |
| | HGERE | 90.2 | 70.7 | 67.5 |

Table 3: F1 scores of Backbone and HGERE with and without a pre-trained span pruner on the SciERC and ACE2005 (BERT$_B$) test set.

| HGERE | SciERC | | |
|---|---|---|---|
| | Ent | Rel | Rel+ |
| Backbone | 71.3 | 52.3 | 40.2 |
| *ter* | 74.2 | 55.1 | 42.6 |
| *sib* | 71.7 | 54.3 | 41.7 |
| *cop* | 71.7 | 52.9 | 40.8 |
| *gp* | 71.3 | 51.9 | 40.1 |
| *tersib* | 74.7 | **55.9** | 43.3 |
| *tercop* | 74.7 | 55.7 | **43.6** |
| *tergp* | 74.5 | 54.9 | 42.4 |
| *tersibcop* | **74.9** | 55.7 | **43.6** |
| *tersibgp* | 74.2 | 54.1 | 41.8 |
| *tercopgp* | 74.7 | 54.0 | 42.3 |
| *tersibcopgp* | 74.3 | 54.6 | 41.7 |

Table 4: F1 scores of HGERE with different graph topologies on the SciERC test set.

and *cop* variants, the NER performance improves slightly, which we attribute to the shared encoder of NER and RE tasks [3]. On the other hand, in the *ter* variant, entity node representations are iteratively refined, resulting in significantly better NER performance than Backbone (74.2 vs. 71.3). Combining *ter* edges with other rel-rel edges (e.g., *sib*) is generally better than using *ter* alone in terms of NER performance, suggesting that joint (and higher-order) modeling of NER and RE indeed has a positive influence on NER, while prior pipeline approaches (e.g., PL-marker) cannot enjoy the benefit of such joint modeling.

For RE, *sib* and *cop* have positive effects on the performance (despite *gp* having a negative effect somehow), showing the advantage of modeling interactions between two different relations. Further combining them with *ter* improves RE performances in all cases, indicating that NER also has a positive effect on RE and confirming again the advantage of joint modeling of NER and RE.

### 5.3 Inference speed of higher-order module

To analyze the computing cost of our higher-order module, we present the inference speed of HGERE with three baseline models Backbone, GCN and MFVI on the test sets of SciERC and ACE2005. Inference speed is measured by the number of candidate entities processed per second. The results are shown in Table 5. We can observe that

---

[3]Though Zhong and Chen (2021) argue that using shared encoders would suffer from the *feature confusion problem*, later works show that shared encoders can still outperform separated encoders (Yan et al., 2021, 2022).

entity spans drops from 99.2 to 78.7 on the SciERC test set, and drops from 99.8 to 95.8 on the ACE2005 test set. We further investigate how the choice of the span pruner vs. the entity identifier influences NER and RE performances. The results are shown in Table 3. We can see that without a span pruner, both NER and RE performances drop significantly, validating the usefulness of using a span pruner. Moreover, it has a consequent influence on the higher-order inference module (i.e., HGNN). Without a span pruner, the improvement from using a HGNN over Backbone is marginal compared to that with a span pruner. We posit that without a pruner many gold entity spans could not exist in the hypergraph of HGNNs, making true entities and relations less connected in the hypergraph and thus diminishing the usefulness of HGNNs.

### 5.2 Effect of the choices of hyperedges

We compare different variants of HGNN with different combinations of hyperedges. Note that if *ter* is not used, entity nodes do not have any hyperedges connecting to them, so their representations would not be refined. We can see that in the *sib*

when utilizing a relatively smaller PLM, `HGERE`, `GCN` and `MFVI` were slightly slower than the first-order model `Backbone`. However, the difference in speed between `HGERE` and the other models was relatively small. When using ALBERT, which is much slower than $\text{BERT}_B$, all four models demonstrated comparable inference speeds.

| | SciERC | ACE2005 | |
| --- | --- | --- | --- |
| | SciBERT | $\text{BERT}_B$ | ALBERT |
| Backbone | 19.4 | 38.0 | 6.1 |
| GCN | 15.7 | 33.8 | 6.3 |
| MFVI | 16.5 | 36.9 | 6.1 |
| HGERE | 15.7 | 30.7 | 6.0 |

Table 5: Comparison of inference speed (#entities/sec) between `HGERE` and three baseline models on test sets of SciERC and ACE2005.

## 5.4 Error correction analysis

We provide quantitative error correction analysis between our higher-order approach `HGERE` and the first-order baseline `Backbone` on the SciERC dataset in Fig. 2. We can see that most error corrections of entities and relations made by `HGERE` come from two categories. The first category is where Backbone incorrectly predicts a true entity or relation as null, and the second category is where Backbone incorrectly assigns a label to a null sample.

## 6 Related Work

**Entity and relation extraction** The entity and relation extraction task has been studied for a long time. The mainstream methods could be divided into pipeline and joint approaches. Pipeline methods tackle the two subtasks, named entity recognition and relation extraction, consecutively (Zelenko et al., 2003; Chan and Roth, 2011; Zhong and Chen, 2021; Ye et al., 2022). By utilizing a new marker-based embedding method, Ye et al. (2022) becomes the new state-of-the-art ERE model. However, pipeline models have the inherent error propagation problem and they could not fully leverage interactions across the two subtasks. Joint approaches, on the other hand, can alleviate the problem by simultaneously tackling the two subtasks, as empirically revealed by Yan et al. (2022). Various joint approaches have been proposed to tackle ERE. Miwa and Bansal (2016); Katiyar and Cardie (2017) use a stacked model for joint learning through shared

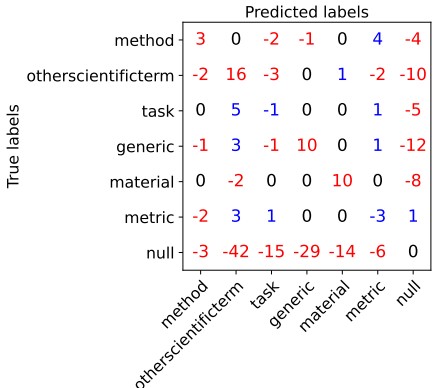

(a) Error correction matrix of `HGERE` vs. Backbone of entities

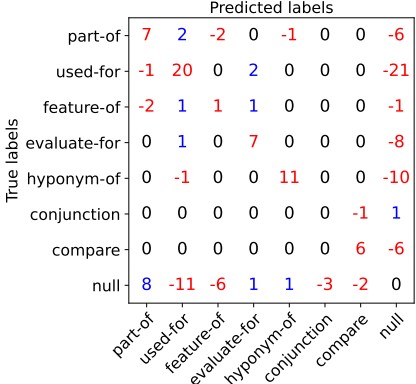

(b) Error correction matrix of `HGERE` vs. Backbone of relations

Figure 2: Error correction of entity and relation types on the SciERC dataset. Red color indicates positive corrections and blue color indicates negative corrections. Specifically, positive numbers on the diagonal of the matrix (in red color) indicate that `HGERE` makes more correct predictions compare to Backbone; negative numbers on non-diagonal entries (in red color) indicate that `HGERE` makes fewer wrong predictions compare to Backbone. Numbers in blue indicate the opposite. We do not count the null-null case.

parameters. Miwa and Sasaki (2014); Gupta et al. (2016); Wang and Lu (2020); Wang et al. (2021); Yan et al. (2021) tackle both the NER and RE tasks as tagging entries of a table. Fu et al. (2019); Sun et al. (2019) leverage a graph convolutional network (GCN) on an instance dependency graph to enhance instance representations. (Nguyen et al., 2021) propose a framework to tackle multiple Information Extraction tasks jointly including the ERE task where a GCN is used to capture the interactions between related instances.

Another line of research is based on text-to-text models for structure prediction including ERE. Normally they are not task-specialized and could solve

several structure prediction tasks in a unified way (Paolini et al., 2021; Lu et al., 2022; Liu et al., 2022).

This work is similar to Sun et al. (2019); Nguyen et al. (2021) for we both use a graph neural network to enhance the instance representations. The main difference is that the GCN they use cannot adequately model higher-order relationship among multiple instances, while our hypergraph neural network is designed for higher-order modeling.

**CRF-based higher-order model**    A commonly used higher-order model utilizes approximate inference algorithms (mean-field variational inference or loopy belief propagation) on CRFs. Zheng et al. (2015b) formulate the mean-field variational inference algorithm on CRFs as a stack of recurrent neural network layers, leading to an end-to-end model for training and inference. Many higher-order models employ this technique for various NLP tasks, such as semantic parsing (Wang et al., 2019; Wang and Tu, 2020) and information extraction (Jia et al., 2022).

**Hypergraph neural network**    Hypergraph neural network (HyperGNN) is another way to construct an higher-order model. Traditional Graph Neural Networks employ pairwise connections among nodes, whereas HyperGNNs use a hypergraph structure for data modeling. Feng et al. (2019) and Bai et al. (2021) proposed spectral-based HyperGNNs utilizing the normalized hypergraph Laplacian. Arya et al. (2020) is a spatial-based HyperGNN which aggregates messages in a two-stage procedure. Huang and Yang (2021) proposed *UniGNN*, a unified framework for interpreting the message passing process in HyperGNN. Gao et al. (2023) introduced a general high-order multi-modal data correlation modeling framework to learn an optimal representation in a single hypergraph based framework.

## 7   Conclusion

In this paper, we present HGERE, a joint entity and relation extraction model equipped with a span pruning mechanism and a higher-order interaction module (i.e., HGNN). We found that simply using the span pruning mechanism by itself greatly improve the performance over prior state-of-the-art PL-marker, indicating the existence of the error propagation problem for pipeline methods. We compared our model with prior tranditional GNN-based models which do not contain hyperedges connecting multiple instances and showed the improvement, suggesting that modeling higher-order interactions between multiple instances is beneficial. Finally, we compared our model with the most popular higher-order CRF models with MFVI and showed the advantages of HGNN in higher-order modeling.

## Limitations

Our model achieves a significant improvement in most cases (on ACE2004, SciERC datasets and on ACE2005 with Bert$_{base}$). While on ACE2005 with stronger encoder (e.g., ALBERT) we observe less siginifical improvements. We posit that, with powerful encoders, the recall of gold entity spans would increase, thereby mitigating the error propagation issue and diminishing the benefit of using a span pruning mechanism.

Another concern regarding our model is computational efficiency. The time complexity of the *Subject-oriented Packing for Span Pair* encoding scheme from PL-marker grows linearly with the size of candidate span size. Recall that we overpredict many spans using a span pruning mechanism, which slows down the running time. In practice, our model's running time is around as three times as that of PL-marker.

## Acknowledgments

This work was supported by the National Natural Science Foundation of China (61976139).

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

# A  Appendix

## A.1  Datasets

We use ACE2004, ACE2005 and SciERC datasets in our experiments, the data statistics of each dataset is shown in Table 6.

|  | #sent | #entity | #relation |
|---|---|---|---|
| ACE2004 | 8683 | 22735 | 4087 |
| ACE2005 | 14525 | 38287 | 7070 |
| SciERC | 2687 | 8094 | 4648 |

Table 6: The statistic of datasets

## A.2  Bidirectional prediction of RE

Following previous work (Eberts and Ulges, 2020; Ye et al., 2022), we establish an inverse relation for each asymmetric relation for a bidirectional prediction. The model can learn the inverse relations of asymmetric relations and improve the performance in this way.

## A.3  Implementation details

We adopt the same cross-sentence information incorporating method used in (Zhong and Chen, 2021; Ye et al., 2022) which extend the original sentence to a fixed window size $W$ with its left and right context. We set $W = 512$ for SciERC, $W = 384$ for ACE2004 and $W = 256$ for ACE2005. For

the pruner training and inference, we consider the span length limitation $L$ of 12 for SciERC and 8 for ACE2004 and ACE2005. For pruners of any datasets and PLMs, the top-K ratio $\lambda = 0.5$, the boundaries of $K$ are $l_{\min} = 3, l_{\max} = 18$. We use three hypergraph convolution layers for GCN, MFVI and HGERE. As the entity recall is high enough, pruners use on ACE2004 and ACE2005 are only trained with $\text{BERT}_B$. For all experiments, we run each configuration with 5 different seeds and report the average micro-F1 scores and standard deviation.

For the pruner, the output sizes of $\text{FFN}_{\text{ST}}, \text{FFN}_{\text{ED}}$ and $\text{FFN}_q$ are $d_m = 768$, the bi-affine embedding size $d_{\text{biaf}} = 256$, the output size of $\text{FFN}_{\text{attn}}$ is 256.

For the backbone module, the output sizes of $\text{FFN}_s, \text{FFN}_o$ and $\text{FFN}_r$ are tuned on $[400, 512, 768]$ for all datasets.

For the hypergraph neural network, the output sizes of $\text{FFN}_r^{ter}, \text{FFN}_s^{ter}, \text{FFN}_o^{ter}, \text{FFN}_a^{z}, \text{FFN}_b^{z}$ are tuned among $[256, 400, 512]$ and fixed on 400 for all experiments on SciERC. The output sizes of $\text{FFN}_e^{ter}, \text{FFN}_e^{z}$ are tune on $[256, 400, 512, 768]$ for all experiments. For GCN, MFVI and HGERE, we all use three layers to refine the node representations. We train our models with Adam optimizer and a liner scheduler with warmup ratio of 0.1. We tune the eps of Adam optimizer on $[1e-8, 1e-9]$ for ACE2005, and eps=$1e-8$ for other datasets. The batch size of all experiments are 18. The learning rate of PLM are $2e-5$, for other module the learning rate is tune on $[5e-5, 1e-4]$. The epochs on SciERC for Backbone are 20, and 30 for other models. The epochs on ACE2004 and ACE2005 ($\text{BERT}_B$) are 15, on ACE2004 and ACE2005 (AL-BERT) are 10. We do all experiments on a A40 GPU with apex fp16 training option on.

### A.4 Details of the span pruner

We obtain contextualized representations of the tokens $\mathbf{x}$ and levitated marker representations $\mathbf{xs}$ (for $[O]$) and $\mathbf{xe}$ (for $[\backslash O]$) . Then we concatenate two kinds of span representations—bi-affine (Dozat and Manning, 2016) and attentive pooling—as the final one. For a span $s_i$ consisting of tokens $x_{\text{ST}(i)}, ..., x_{\text{ED}(i)}$, its bi-affine span representation

is a $d_{\text{biaf}}$-dimension vector,

$$\mathbf{h}_{\text{ST}}(s_i) = \text{FFN}_{\text{ST}}(\mathbf{x}_{\text{ST}(i)}; \mathbf{xs}(i))$$
$$\mathbf{h}_{\text{ED}}(s_i) = \text{FFN}_{\text{ED}}(\mathbf{x}_{\text{ED}(i)}; \mathbf{xe}(i))$$
$$\mathbf{h}_{\text{biaf}}(s_i) = [\mathbf{h}_{\text{ST}}(s_i); 1]^\top W_p [\mathbf{h}_{\text{ED}}(s_i); 1]$$

the symbol ; is the concatenation operation, $\text{FFN}_{\text{ST}}$ and $\text{FFN}_{\text{ED}}$ are feed-forward layers with an output size $d_m$ and $W_p \in \mathbb{R}^{(d_m+1)*d_{\text{biaf}}*(d_m+1)}$ is a learn-able weight. The attentive pooling layer is a weighted average over the contextualize token representations in the span,

$$w_j = \text{FFN}_q(\mathbf{x}_j); w_j = \frac{\exp w_j}{\sum_{\text{ST}(i) \leq l \leq \text{ED}(i)} \exp w_l}$$
$$\mathbf{h}_{\text{attn}}(s_i) = \sum_{\text{ST}(i) \leq j \leq \text{ED}(i)} w_j \mathbf{x}_j$$

and the final span representation is,

$$\mathbf{h}(s_i) = \text{FFN}_{\text{attn}}[\mathbf{h}_{\text{biaf}}(s_i); \mathbf{h}_{\text{attn}}(s_i)]$$

**Training and Inference** Given the gold binary tag $y(s_i) \in \{0, 1\}$ (indicating the existence of a candidate span in the gold span set), we train the span pruner with the binary cross-entropy (BCE) loss:

$$\hat{y}(s_i) = \text{Sigmoid}(\text{FFN}(\mathbf{h}(s_i)))$$

$$L = -\sum_{1 \leq i \leq m} [y(s_i) \log \hat{y}(s_i) \\ + (1 - y(s_i))(1 - \log \hat{y}(s_i)]$$

### A.5 Mean-field Variant Inference

Here we introduce the method used in baseline MFVI. The hyperedges in our graph are replaced by factors in MFVI, so there are also four kinds of factors: *ter*, *sib*, *cop*, *gp*.

**first-order scores** We use the node representations to score the entities and relations for each label (include the null).

$$\mathbf{u}_i^s = \text{FFN}_s^u(\mathbf{g}(v_s^i))$$
$$\mathbf{u}_j^o = \text{FFN}_o^u(\mathbf{g}(v_o^j))$$
$$\mathbf{u}_{ij}^r = \text{FFN}_r^u(\mathbf{g}(v_r^{ij}))$$

$\mathbf{u}^s, \mathbf{u}^o \in R^{|\mathcal{C}_e|+1}, \mathbf{u}^r \in R^{|\mathcal{C}_r|+1}$.

**Higher-order scores**   Each factor scores the joint distribution of the node types connected to it. For a *ter* factor connects a subject $v_s^i$, an object $v_o^j$ and a relation $v_r^{ij}$, the factor score $\mathbf{f}_{ij}^{ter} \in R^{|\mathcal{C}_e+1|^2|\mathcal{C}_r+1|}$ is:

$$\mathbf{h}_i^s = \text{FFN}_s^{ter}(\mathbf{g}(v_s^i))$$
$$\mathbf{h}_j^o = \text{FFN}_o^{ter}(\mathbf{g}(v_o^j))$$
$$\mathbf{h}_{ij}^r = \text{FFN}_r^{ter}(\mathbf{g}(v_r^{ij}))$$
$$\mathbf{f}_{ij}^{ter} = \text{FFN}_f^{ter}(\mathbf{h}_i^s \circ \mathbf{h}_j^o \circ \mathbf{h}_{ij}^r)$$

For a factor z, $z \in \{sib, cop, gp\}$ which connects two relations, we name them relation $a$ and $b$ for simplicity. If relation $a$ is $v_r^{ij}$, then relation $b$ is $v_r^{ik}, v_r^{kj}$ and $v_r^{jk}$ for *sib*, *cop* and *gp* respectively. We use $\mathbf{g}(a), \mathbf{g}(b)$ to refer to the relation representations of relations $a$ and $b$. The factor score $\mathbf{f}_{ijk}^z \in R^{|\mathcal{C}_r+1|^2}$ is defined as:

$$\mathbf{h}_a = \text{FFN}_s^{ter}(\mathbf{g}(a))$$
$$\mathbf{h}_b = \text{FFN}_o^{ter}(\mathbf{g}(b))$$
$$\mathbf{f}_{ijk}^z = \text{FFN}_f^{ter}(\mathbf{h}_a \circ \mathbf{h}_b)$$

**higher-order inference**   In the model, computing the node distribution can be seen as doing posterior inference on a Conditional Random Field (CRF). MFVI iteratively updates a factorized variational distribution Q to approximate the posterior label distribution. We use $Q_{s_i}(e_1), Q_{o_j}(e_2)$ to refer to the probability of subject $v_s^i$ and object $v_o^j$ has entity type $e_1$ and $e_2$ respectively and $Q_{r_{ij}}(r)$ represents the relation $v_r^{ij}$ has the relation type $r$. For simplicity, we use $u_i^s(e_1), u_j^o(e_2), u_{ij}^r(r_1), f_{ij}^{ter}(e_1, e_2, r_1), f_{ijk}^z(r_1, r_2)$ to represent the first-order and higher-order scores when the subject $v_s^i$, the object $v_o^j$ have entity type $e_1, e_2$, the relation $a$ ($v_r^{ij}$), the relation $b$ have relation types $r_1, r_2$ respectively. Following is the iterately updating of the distribution $Q_s, Q_o, Q_r$. For a subject $v_s^i$. The message only passed from *ter* factor in the $l$-th iteration is:

$$F_{s_i}^l(e_1) =$$
$$\sum_j \sum_{e_2} Q_{o_j}^{l-1}(e_2)(\sum_{l_1} Q_{r_{ij}}^{l-1}(r_1)f_{ij}^{ter}(e_1, e_2, r_1))$$

similarly, the message passed from *ter* factor to the object $v_o^j$ is:

$$F_{o_j}^l(e_2) =$$
$$\sum_i \sum_{e_1} Q_{s_i}^{l-1}(e_1)(\sum_{l_1} Q_{r_{ij}}^{l-1}(r_1)f_{ij}^{ter}(e_1, e_2, r_1))$$

For a relation $v_r^{ij}$, the message could be passed from four factors, we list them by the source. From *ter* factor:

$$T_{r_{ij}}^l(r_1) =$$
$$\sum_{e_1} \sum_{e_2} Q_{s_i}^{l-1}(e_1)Q_{o_j}^{l-1}(e_2)f_{ij}^{ter}(e_1, e_2, r_1)$$

From *sib* factor:

$$S_{r_{ij}}^l(r_1) =$$
$$\sum_k \sum_{r_2} Q_{r_{ik}}^{l-1}(r_2)(f_{ijk}^{sib}(r_1, r_2) + f_{ikj}^{sib}(r_2, r_1))$$

From the *cop* factor:

$$C_{r_{ij}}^l(r_1) =$$
$$\sum_k \sum_{r_2} Q_{r_{kj}}^{l-1}(r_2)(f_{ijk}^{cop}(r_1, r_2) + f_{kji}^{cop}(r_2, r_1))$$

From the *gp* factor:

$$G_{r_{ij}}^l(r_1) =$$
$$\sum_k \sum_{r_2} (Q_{r_{jk}}^{l-1}(r_2)f_{ijk}^{gp}(r_1, r_2)$$
$$+ Q_{r_{ki}}^{l-1}(r_2)f_{kij}^{gp}(r_2, r_1))$$

The posterior distribution of entity $e_i$ with respect to the subject $s_i$ and object $o_i$ :

$$Q_{s_i}^l(e) \propto \exp(u_i^s(e) + F_{s_i}^l(e))$$
$$Q_{o_j}^l(e) \propto \exp(u_j^o(e) + F_{o_j}^l(e))$$

Then the entity distribution is : $Q_{s_i}^l + Q_{o_i}^l$

We initial the Q of subject $v_i^s$, object $v_j^o$, the relation $v_{ij}^r$ by normalizing the unary potential $\exp(u_i^s), \exp(u_j^o), \exp(u_{ij}^r)$ respectively. The posterior distribution of the relation $r_{ij}$ is:

$$Q_{r_{ij}}^l(r) \propto \exp(u_{ij}^r(r) + \mathbb{1}_{ter}T_{r_{ij}}^l(r)$$
$$\mathbb{1}_{sib}S_{r_{ij}}^l(r) + \mathbb{1}_{cop}C_{r_{ij}}^l(r) + \mathbb{1}_{gp}G_{r_{ij}}^l(r))$$

The symbol $\mathbb{1}_z$, $z \in \{ter, sib, cop, gp\}$ indicates whether the factor z exists in the graph.

## A.6   GCN

Here is the introduction of the baseline GCN. As in HGERE, we also build the graph $\mathcal{G} = (\mathcal{V}, \mathcal{E})$ with subject, object and relation nodes, $\mathcal{V} = \mathcal{V}_s \bigcup \mathcal{V}_o \bigcup \mathcal{V}_r$. For each relation node $v_r^{ij} \in \mathcal{V}_r$, we build two edges connecting its subject node $v_s^i \in \mathcal{V}_s$ and object node $v_o^j \in \mathcal{V}_o$ respectively. the

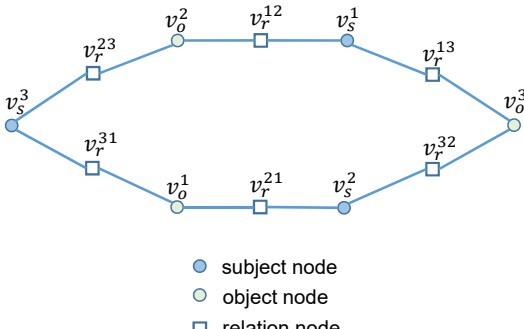

subject node
object node
relation node

Figure 3: Illustration of an example graph of GCN

model use $l$ convolution layers to update the node representations. We define the neighbor set $\mathcal{N}(v)$ of a node $v$ is the nodes connected to it. The node representation update of $l$-th layer is as follow:

$$\beta^l(v_1, v) = \mathbf{w}^\top \sigma(\mathbf{W}[\mathbf{g}^{l-1}(v_1); \mathbf{m}^{l-1}(v)])$$

$$\alpha^l(v_1, v) = \frac{\exp \beta^l(v_1, v)}{\sum_{v_1 \in \mathcal{N}(v)} \exp \beta^l(v_1, v)}$$

$$\mathbf{g}^l(v) = \mathbf{g}^{l-1}(v) + \sum_{v_1 \in \mathcal{N}(v)} \alpha(v_1, v)\mathbf{g}^{l-1}(v_1)$$

### A.7 Performance with part of the training data

| ratio | model | ACE2005 (BERT$_B$) | | |
|---|---|---|---|---|
| | | Ent | Rel | Rel+ |
| 5% | Backbone | 80.4 | 39.7 | 36.0 |
| | HGERE | 79.5 | 42.0 | 38.1 |
| 10% | Backbone | 83.9 | 51.3 | 47.2 |
| | HGERE | 84.2 | 53.3 | 49.4 |
| 100% | Backbone | 90.0 | 69.8 | 66.7 |
| | HGERE | 90.2 | 70.1 | 67.3 |

Table 7: F1 score of HGERE on ACE2005 test set when only provide 5% and 10% training samples.

From the main results, we can see that the HGERE shows a significantly greater improvement in performance compared to the Backbone model on the SciERC dataset than on the ACE2005 dataset. We guess one of the reason is the size of the training data. Because with more training data, models could learn enough knowledge from a large number of samples and reduce the demand of higher-order information. So we compare HGERE to Backbone with 5% and 10% of training data on the ACE2005 (BERT$_B$) to see if higher-order inference is more effectiveness with small training data. From the

| | SciERC | | |
|---|---|---|---|
| | Ent | Rel | Rel+ |
| max | 74.0 | 54.7 | 41.4 |
| sum | 73.6 | 54.5 | 41.5 |
| attn | 74.9 | 55.7 | 43.6 |

Table 8: F1 scores of HGERE (the *tersibcop* variant) with different aggregation functions on the SciERC test set.

results shown in Table 7 we can see that the increments of absolute F1 score on Rel+ metric from Backbone to HGERE are 2.1%, 2.2% on 5% and 10% of training set respectively, which are much higher than 0.6% on full training set.

### A.8 Effect of the number of HGNN layers

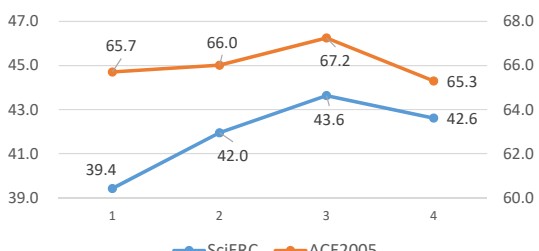

Figure 4: The change of F1 scores with respect to the number of HGNN layers .

From Fig.4 we can see that using three HGNN layers performs the best while more layers lead to worse results. We posit that this is because using more HGNN layers would suffer from the well-known over-smoothing problem (Cai and Wang, 2020).

### A.9 Effect of the aggregation function in message passing

We study the influence of using different message aggregation functions. HGERE uses an attention mechanism (attn) to update node representations while it is also possible to use max-pooling (max) or sum-pooling (sum). Table 8 shows that attn performs the best.