# OpenReview forum: "Joint Entity and Relation Extraction with Span Pruning and Hypergraph Neural Networks"
_EMNLP/2023/Conference — EMNLP 2023 Main_

### Official Review · Reviewer_tBFS · 2023-08-02

**Soundness:** 4

**Excitement:**

3: Ambivalent: It has merits (e.g., it reports state-of-the-art results, the idea is nice), but there are key weaknesses (e.g., it describes incremental work), and it can significantly benefit from another round of revision. However, I won't object to accepting it if my co-reviewers champion it.

**Missing References:**

It would be better to discuss related work on learning with hypergraphs.
[nips06]Learning with Hypergraphs: Clustering, Classification, and Embedding
[neurips21]Edge Representation Learning with Hypergraphs

**Paper Topic And Main Contributions:**

The paper develops an entity relation extraction model based on the LP-marker (which is derived from a pipeline extractor PURE). The key technical contribution is to make connections among entities and relations closer. Specifically, The method includes more candidates in the first entity extraction step and deploys a stronger relation extractor (though GNN with hypergraph) to determine which entities will finally be extracted (by consulting relation signals).

**Reasons To Accept:**

- the paper is well-written and easy to follow.
- the results are strong.


**Reasons To Reject:**

- though the model gets a strong performance, both using levitated markers and using graph neural networks for high-order interaction have been studied in previous tasks.
- It seems that the hypergraphs are fully connected in the sense that no weights are assigned on the edges. it would be better to study other ways to build the graph (e.g., pruning some edges, even using ordinary graphs rather than hypergraphs).


**Reproducibility:**

4: Could mostly reproduce the results, but there may be some variation because of sample variance or minor variations in their interpretation of the protocol or method.

**Reviewer Confidence:**

4: Quite sure. I tried to check the important points carefully. It's unlikely, though conceivable, that I missed something that should affect my ratings.

---

> ### Author Rebuttal · Authors · 2023-08-29
>
> Thank you for your useful comments. To provide a better understanding of the significance of our work, let us first define higher-order models.
>
> In our paper, **first-order** models refer to models that only utilize the features or distributions of individual variables. GNN models with a traditional graph backbone like GCN can be considered as **second-order** models since they utilize pairwise relationships. In contrast, our HGNN-based HGERE model captures features of hyperedges connecting three nodes (ternary edge in the paper), making it a **third-order** model. Similarly, CRFIE[1], which published in ACL2023,defines ternary factors connecting three nodes and is therefore also a third-order model. Note that some previous studies using GCN also claim to be higher-order models by arguing that one GCN layer encodes information about immediate neighbors and K layers encode Kth-order neighborhoods. However, according to our definition, they can only be considered as second-order models.
> >[1]: Modeling Instance Interactions for Joint Information Extraction with Neural High-Order Conditional Random Field
>
>
>
> ## Regarding using GNN in previous study
>
> Note that HGNNs are different from and more general than GNNs. GNNs are only capable of second-order modeling, while HGNNs are capable of higher-order modeling. We are the first to utilize HGNN for higher-order ERE.
>
> ## Regarding levitated markers in previous study
>
> We simply build our higher-order model upon a marker-based model and levitated marker itself is not our contribution.
>
> ## Regarding the question about "...hypergraphs are fully connected in the sense that no weights are assigned on the edges."
>
> The hypergraph in the paper is not fully connected as that would result in very high complexity. We considered only two types of manually designed hyper-edges: *sub-obj-rel* and *rel-rel*. This is the most natural way for higher-order modeling of ERE as it connects potentially related nodes using hyperedges. Ordinary graphs are only capable of second-order modeling and cannot capture higher-order relationships such as *sub-obj-rel*.

---

### Official Review · Reviewer_YBzL · 2023-08-04

**Soundness:** 3

**Excitement:**

3: Ambivalent: It has merits (e.g., it reports state-of-the-art results, the idea is nice), but there are key weaknesses (e.g., it describes incremental work), and it can significantly benefit from another round of revision. However, I won't object to accepting it if my co-reviewers champion it.

**Missing References:**

[1] HIORE: Leveraging High-order Interactions for Unified Entity Relation Extraction

**Paper Topic And Main Contributions:**

This paper proposes HGERE, a joint entity and relation extraction model equipped with a span pruning mechanism and a higher-order interaction module.
It is built upon the state-of-the-art PL-marker, and achieves higher-order inference with hypergraph neural networks.
Experiments on 3 datasets show the effectivesss.

**Reasons To Accept:**

1. Paper is well-written and the idea is intuitive.
2. The results are good.

**Reasons To Reject:**

1. There has been similar work [1] on GNN for capturing higher-order interactions. It weakens the contribution of this paper.
2. This IE research paradigm is a bit outdated in the age of LLMs.

**Reproducibility:**

4: Could mostly reproduce the results, but there may be some variation because of sample variance or minor variations in their interpretation of the protocol or method.

**Reviewer Confidence:**

4: Quite sure. I tried to check the important points carefully. It's unlikely, though conceivable, that I missed something that should affect my ratings.

---

> ### Author Rebuttal · Authors · 2023-08-29
>
> Thank you for reviewing our work.
>
> 1. HIORE [1] is a preprint paper published in May 2023 and is therefore concurrent to our work.
>
> 2. While it is true that LLM-based IE is becoming more popular, it is important to embrace a diverse range of research directions and not focus on a single type of methodology.
> There is also evidence showing that with sufficient training data, LLM-based IE methods still underperform traditional encoder-based IE methods.

---

### Official Review · Reviewer_g3Cy · 2023-08-11

**Soundness:** 3

**Excitement:**

3: Ambivalent: It has merits (e.g., it reports state-of-the-art results, the idea is nice), but there are key weaknesses (e.g., it describes incremental work), and it can significantly benefit from another round of revision. However, I won't object to accepting it if my co-reviewers champion it.

**Paper Topic And Main Contributions:**

This paper proposes a joint entity and relation extraction model, utilizing span pruning and GNN to make improvements over SOTA approach, PL-maker. The paper analyzes the effects of span pruning and HGNN separately. The results on three dataset shows the improvement.

Weakness: Lack of innovation. The span pruning method is simple and there are some related works. There are also some related works based on higher-order modeling approaches. It is interesting to use SOTA GNN to replace MFVI but the innovation is limited. If you can make a deep analysis on more different SOTA higher-order models or graph neural networks, it will be better and more convincing.

**Questions For The Authors:**

- Is this the first work that explicit graph neural network on the ERE task?

- Beside HyperGNN and GCN, do you try other SOTA graph network modeling method?

- Is there any other SOTA higher-order modeling approach baseline? MFVI seems too old…

- Line 447. Report the Precision and Recall to make a clear comparison and analysis.

- Could you report the computing costs.

- Line 430. Is there any experiments to show the span pruning reduce the error propagation?

**Reasons To Accept:**

The application of span pruning and Hyperbolic Graph Neural Networks (HGNN) for addressing the ERE task demonstrates good performance. It is interesting to see the improvement over PL-maker, and this works analyzes the improvement brought from GNN.

**Reasons To Reject:**

Utilizing state-of-the-art Graph Neural Networks (GNNs) as a higher-order modeling technique is intriguing; however, the scope of innovation appears constrained. The work consider GCN and MFVI as baseline and shows the effect of higher-order model but the improvement seems limited. Also the paper should add more higher-order baselines.

**Reproducibility:**

5: Could easily reproduce the results.

**Reviewer Confidence:**

4: Quite sure. I tried to check the important points carefully. It's unlikely, though conceivable, that I missed something that should affect my ratings.

---

> ### Author Rebuttal · Authors · 2023-08-29
>
> Thank you for your valuable feedback and questions regarding our work. To provide a better understanding of the significance of our work, let us first define higher-order models.
>
> In our paper, **first-order** models refer to models that only utilize the features or distributions of individual variables. GNN models with a traditional graph backbone like GCN can be considered as **second-order** models since they utilize pairwise relationships. In contrast, our HGNN-based HGERE model captures features of hyperedges connecting three nodes (ternary edge in the paper), making it a **third-order** model. Similarly, CRFIE[1], which published in ACL2023,defines ternary factors connecting three nodes and is therefore also a third-order model. Note that some previous studies using GCN also claim to be higher-order models by arguing that one GCN layer encodes information about immediate neighbors and K layers encode Kth-order neighborhoods. However, according to our definition, they can only be considered as second-order models.
> >[1]: Modeling Instance Interactions for Joint Information Extraction with Neural High-Order Conditional Random Field
>
>
>
> ## Regarding the innovation
> 1. As far as we know, CRFIE is the only previous approach to higher-order Entity and Relation Extraction, which is based on CRF+MFVI. We are the first to utilize HGNNs for higher-order modeling. The advantages of HGNNs over CRF+MFVI are discussed in the paragraph starting at line 098.
> 2. In the field of IE, span-based models generally outperform token-based models but have higher computational complexity (especially for higher-order modeling). To address this problem, most previous span-based models employed entity identification as an initial step to filter entities. However, a major drawback of this approach is that errors from the first step propagate to downstream models, impacting the performance of the models. In contrast, our span pruner not only reduces the complexity of higher-order models but also significantly improves their performance (as demonstrated in Section 5.1). Although span pruning has been researched previously, it is novel to combine it with HGNNs in higher-order span-based ERE, improving both accuracy and efficiency.
>
> ## Answering the questions
>
> 1. We are the first to utilize **HGNN** for ERE. Note that HGNNs are different from and more general than GNNs.
>
> 2. We have tried GAT but it achieves similar performance compared to the GCN baseline in our paper. Meanwhile the state-of-the-art ERE model PL-marker, does not use GNN and achieves superior performance compared to previous GNN-based models. So in additional to PL-marker, we choose current baselines in the paper.
>
> 3. As mentioned above, there is limited research on higher-order models and there are no higher-order models available for comparison other than third-order MFVI.
>
> 4-5. We will report them in the final edition.
>
> 6. The experiments in Section 5.1 demonstrate the performance differences between models with an entity pruner vs. an entity identifier, which verify how the pruner reduces error propagation.

---

### Meta-Review · Area_Chair_2iyw · 2023-09-20

**Recommendation:** 3

**Metareview:**

This paper proposes HyperGraph neural network for joint entity and relation extraction to alleviate the error propagation in marker-based pipeline models, and capture higher-order interactions between multiple entities and relations. For all the reviewers, the most concern is its novelty and contribution. Comparing with prior methods, there are indeed something new in the proposed method. It would be nice if the authors could make precise revisions according to their responses.

---

### Decision · Program_Chairs · 2023-10-07

**Decision:**

Accept-Main

**Comment:**

This paper proposes HyperGraph neural network for joint entity and relation extraction to alleviate the error propagation in marker-based pipeline models, and capture higher-order interactions between multiple entities and relations. For all the reviewers, the most concern is its novelty and contribution. Comparing with prior methods, there are indeed something new in the proposed method. It would be nice if the authors could make precise revisions according to their responses.